# Two distinct ship emission profiles for organic-sulfate source apportionment of PM in sulfur emission control areas

Kirsten N. Fossum[1], Chunshui Lin[1,2,3], Niall O'Sullivan[4], Lu Lei[1], Stig Hellebust[4], Darius Ceburnis[1], Aqeel Afzal[1,5], Anja Tremper[6], David Green[6,7], Srishti Jain[4], Steigvilė Byčenkienė[8], Colin O'Dowd[1], John Wenger[4], and Jurgita Ovadnevaite[1]

[1]School of Natural Sciences, Ryan Institute's Centre for Climate & Air Pollution Studies, University of Galway, Galway, H91 CF50, Ireland
[2]State Key Laboratory of Loess and Quaternary Geology and Key Laboratory of Aerosol Chemistry and Physics, Chinese Academy of Sciences, 710061, Xi'an, China
[3]Department of Civil and Environmental Engineering, the Hong Kong Polytechnic University, Hong Kong, China
[4]School of Chemistry and Environmental Research Institute, University College Cork, Cork, Ireland
[5]Institute of Energy and Environmental Engineering, University of the Punjab, Lahore, Pakistan
[6]MRC Centre for Environment and Health, Environmental Research Group, Imperial College London, UK
[7]NIHR HPRU in Environmental Exposures and Health, Imperial College London, UK
[8]SRI Center for Physical Sciences and Technology, Lithuania

*Correspondence to*: Jurgita Ovadnevaite (jurgita.ovadnevaite@universityofgalway.ie)

**Abstract.** Source apportionment quantitatively links pollution to its source, but can be difficult to perform in areas like ports where emissions from shipping and other port-related activities are intrinsically linked. Here we present the analysis of aerosol chemical speciation monitor (ACSM) data using combined organic and sulfate ion positive matrix factorization (PMF) during an intensive measurement campaign in Dublin Port. Two main types of ship emissions were identified by this technique: sulfate-rich (S-Ship) and organic-rich (O-Ship). The S-Ship emissions were attributed to heavy fuel oil use and are characterised by particles with standard V/Ni ratios from 2.7-3.9 and a large fraction of acidic sulfate aerosol. The O-Ship emissions were attributed to low-sulfur fuel types and were comprised mostly of organic aerosol (OA) with the V/Ni ratios ranging only from 0-2.3. O-Ship plumes occurred over three-times more frequently than S-Ship plumes during the measurement period. A third minor ship emission factor (X-Ship) was resolved by PMF, but not clearly attributable to any specific fuel type. Overall, ship plumes had $PM_1$ concentrations in the range 4-252 µg m$^{-3}$, with extreme concentrations usually lasting for 5-35 minutes. Despite their short duration, shipping emission plumes were frequent and contributed to at least 28% of $PM_1$ (i.e. 14% O-Ship, 12% S-Ship, and 2% X-Ship). Moreover, hydrocarbon-like organic aerosol (HOA) and black carbon could also originate, in part, from shipping related activities such as ferry traffic and heavy goods vehicles, suggesting that the overall contribution of shipping activity to ambient $PM_1$ is likely higher, upwards of 47%.

## 1 Introduction

Shipping traffic is set to expand worldwide, increasing pollution in port areas and potentially leading to poorer air quality for 37% of the world's population living within 100 km of the coast (UN Environment Programme, 2024). A range of emission sources influence the air quality in port areas, including combustion sources such as ocean-going vessels, heavy goods vehicles and land-based industry. These emissions have many similar chemical components, and it can be difficult to separate individual sources, especially when they may be intrinsically related e.g., primary ship emissions and secondary formation of aerosol from ship related precursor gases. However, a combination of chemical analysis methods and source apportionment modelling can be used to successfully determine the contribution of specific sources to the ambient particulate matter measured in port areas. For example: 3.7–6.1% of organic aerosol was related to shipping and industrial plumes in Marseille, France (Chazeau et al., 2022), 1.5% of $PM_{2.5}$ and 18% of particle number concentration was related to shipping traffic in Cork Harbor, Ireland (O'Connor et al., 2013; Healy et al., 2010), shipping emissions were 5–14% of $PM_{2.5}$ in Spanish coasts (Pandolfi et al., 2011; Viana et al., 2009) and were 4–13% of primary $PM_{2.5}$ in Shanghai Port and Hong Kong Ports (Yau et al., 2013; Zhao et al., 2013) and 25% overall in Hong Kong Port (Yau et al., 2013). In Ningbo-Zhoushan Port, China, 18% of polycyclic aromatic hydrocarbons (PAHs) in $PM_{2.5}$ were found to come from Heavy Fuel Oil (HFO) combustion (Hong et al., 2023), and shipping emissions contributed 6–22% of volatile organic compounds in the Pearl River Delta region (Tong et al., 2024).

High time resolution measurements of aerosol chemical composition can be used to identify different emission sources in port areas, as they are capable of reflecting the transient emission sources and changing meteorology. For example,

Vanadium (V) and Nickel (Ni) have been used as chemical tracers to identify primary emissions from combustion of HFO (Healy et al., 2009; Mueller et al., 2011; Agrawal et al., 2009; Zhao et al., 2013) and concentration ratios of V/Ni ranging from 2.5–4.0 are associated with typical ship emissions (Mazzei et al., 2008; Pandolfi et al., 2011; Viana et al., 2009). Different analysis techniques can be used to perform source apportionment; one leading type of multivariate analysis for high-resolution aerosol composition is positive matrix factorisation (PMF), which is capable of resolving distinct primary as well as secondary aerosol sources (*e.g.* Chazeau et al., 2022; Yau et al., 2013).

There are many regulations and guidelines related to the control of emissions and air quality in port areas. Among these are sulfur emission control areas (SECA), which aim to reduce emissions of sulfur oxides from ships by limiting the sulfur-content in marine fuel. These regulations aim to improve health and reduce negative ecosystem impacts of commercial shipping and are enforced in Europe through the EU Sulfur Directive and at the international level by the International Maritime Organisation (IMO). On January 1[st], 2015, the IMO reduced the limit on sulfur fuel content in SECA from 1.0% m/m (mass by mass) to 0.1% m/m. Additionally, the maximum sulfur content outside of SECA was reduced from 3.5% m/m to 0.5% m/m on January 1[st], 2020 (IMO 2020). Due to the higher cost of low-sulfur (low-S) fuels, many ship operators have instead installed exhaust scrubber systems, which reduce the gaseous sulfur emissions. A common wet scrubber design uses alkaline solution, often seawater pumped from below the ship, to spray through the ship exhaust, scavenge, and reduce gaseous $SO_2$ emissions. Vessels with exhaust scrubber systems, in accordance with an amendment to the original regulations, are still allowed to use fuels exceeding 0.5% sulfur after March 1[st] 2020. This has many implications for both the composition of the aerosol emissions and for seawater acidification and pollution (Comer et al. 2020). The transition from HFO (S<3.5% m/m) to ultra-low-S fuel (0.1% m/m) has been shown to improve air quality through reduced mass concentration of particulate matter (PM) by 67%, reduced $SO_2$ emissions by 80% and an overall decrease in volatile organic compounds, including the heavier and carcinogenic PAHs, (Zetterdahl et al., 2016). However, this transition has also been shown to increase the production of monoaromatic and lighter polyaromatic hydrocarbon compounds (Zetterdahl et al., 2016). Despite reductions in many pollutants, the lowering of the sulfur fuel content is unlikely to lead to significant changes in either the total particle number concentration or the black carbon mass concentration (Zetterdahl et al., 2016). Studies have pointed out that low-S fuels contain much lower amounts of metals from the refinery process and therefore will not have the typical chemical markers of HFO traditionally used for tracing ship emissions (Anders et al., 2023; Czech et al., 2017). While it has been proposed that lubricant oil from marine engines could provide a fuel-independent pool of possible marker substances (Eichler et al., 2017), new studies are urgently needed in port areas to derive alternative markers or chemical profiles for ship emissions, as well as diagnostic ratios for both particle-bound and volatile organics (Czech et al., 2017).

Dublin Port is the largest port in Ireland, classified as a Tier 1 medium port. In 2019, it handled 49.5% (~26.3 million tonnes) of Irish Freight (Transport Omnibus (2019). For context, the largest port in the EU, Rotterdam Port, has 18-times this capacity. Dublin Port expects to double its capacity by 2040, at a 3.3% expansion rate per annum (Dublin Port Masterplan 2040 (DPC, 2018)). Dublin Port is adjacent to the urban centre of Dublin city (< 5 km), where the air quality has been

studied at both urban background and roadside monitoring locations (Lin et al. 2018, Lin et al. 2019, Ovadnevaite et al. 2021). Dublin is known to be diurnally affected by poor air quality arising from the burning of domestic solid fuels for home heating during the colder (mainly winter) months, often with night-time peaks exceeding 100 µg m$^{-3}$ (sometimes > 300 µg m$^{-3}$) for several hours (Lin et al. 2018, Ovadnevaite et al. 2021). Dublin Port is directly downwind of the prevailing Westerlies (South-Westerlies) and as such is impacted by both the air pollution from the Port and the City Centre of Dublin. As Dublin Port is a SECA, ships either switch to ultra-low-S content fuels while at dock or else implement the use of scrubbers aboard the ship to reduce SO$_2$ emissions from burning fuels with higher S content. The resultant particulate emissions from the use of scrubbers would have the same V/Ni signatures of HFO, while supporting the rapid aqueous phase formation of acidic sulfate (SO$_4^{2-}$) within the plume stacks. In fact, studies of before and after scrubber system installation confirm the presence of SO$_4^{2-}$ in the aerosol particle phase from ship stacks with scrubbers (Yang et al., 2021). Conversely, ultra-low-S fuels (S < 0.1% m/m) as well as Very Low Sulfur Fuel Oil (VLSFO, S < 0.5% m/m) lack the processing that yield metal tracers (V/Ni) from the combustion of the fuel. The most common fuel use behaviours at Dublin Port were (i) using ultra-low-S fuels only (mainly Marine Gas Oil (MGO)), (ii) using VLSFO to power the engines and MGO for electricity generators when in port, (iii) using HFO for engines (with scrubber) and MGO for generators when in port, and (iv) using HFO with a wet scrubber operated using a closed loop system all the time.

A research project, *Source Apportionment of Air Pollution in the Dublin Port Area* (PortAIR), was initiated to measure the aerosol physical and chemical properties in the port area and assess the impact of Dublin Port activities on air quality before it doubles capacity by 2040. The PortAIR project comprises a 14-month long air quality field campaign (December 2021 – February 2023) and an 8-week long intensive measurement campaign (December 2022 – February 2023) at a monitoring site in Dublin Port, situated ~5 km from the city centre. Here we present results from a 1-month period of the intensive campaign conducted in winter, when air quality was affected both by burning of domestic solid fuels in the city and by peak port activity from goods importation. The comprehensive range of instruments deployed at the monitoring site allowed characterisation of individual ship plumes and classification according to type of fuel used.

## 2 Methods

### 2.1 Measurement campaign

This study focuses on an intensive field measurement campaign in Dublin Port where aerosol physico-chemical properties and gaseous pollutants were measured using a suite of instrumentation housed in two containers. The intensive campaign ran from 16 December 2022 through to 7 February 2023. The monitoring site (latitude of 53.348439 and longitude of -6.194657) was selected to be downwind of most port activity and close to the ferries, which are a major daily source of shipping emissions. The location of the monitoring site in relation to the ferry terminals and other areas of the port is shown in Supplementary Fig. S1, along with a photograph of the two containers *in situ*.

High time resolution chemical composition data from a monitoring site around 5 km from Dublin Port is also used in this study. The site is at University College Dublin (UCD) (53.3089, -6.2242), an urban background location just South of Dublin city centre, close to main roads and residential areas (Lin et al. 2020; Lin et al. 2018).

## 2.2 Instrumentation

### 2.2.1 Meteorology

Wind direction, wind speed, air temperature, air pressure, relative humidity (RH), rainfall and solar radiation measurements were made using a *Casella* weather station (model Nomad, UK) mounted to the top of the main container. The measured wind speed and direction compared well with the data available from the nearest Met Éireann meteorological station located at Dublin Airport, less than 10 km North of Dublin Port.

The wind speed and wind direction were compiled into the Igor software compatible tool for geographical origins of atmospheric pollution, ZeFIR (Petit et al., 2017), to plot air pollution roses aided by the openair package (v2.8-3; Carslaw and Ropkins, 2012).

### 2.2.2 Q-ACSM

A PM$_1$ quadrupole aerosol chemical speciation monitor (Q-ACSM) from *Aerodyne Inc.* (Billerica, MA, USA) measured the
mass concentrations of non-refractory species including organic aerosol (OA), sulfate (SO$_4^{2-}$), nitrate (NO$_3^-$), ammonium (NH$_4^+$) and chloride (Cl$^-$) (Ng et al., 2011). While the intensive campaign ran from 16 December 2022 to 7 February 2023, Q-ACSM data is only available through to January 27, 2023. The Q-ACSM used in the study had a standard vaporiser and was calibrated and maintained following the standard protocol developed by the Cost Action CA16109, COLOSSAL. Details of the Q-ACSM instrument can be found in previous studies (e.g. Ng et al. (2011) and Pieber et al. (2016)). In this
study, the Q-ACSM was installed with a PM$_{2.5}$ URG-2000-30ED cyclone connected to 3/8 inch stainless steel tubing and operated using a carrier flow rate of 2.5 ($\pm$0.2) LPM with a distance from inlet to Q-ACSM of approximately 2 m, to keep particle losses to a minimum. A monotube Nafion® membrane dryer was installed to maintain RH of the sample air in the range 20–40%. The instrument was operated at a time resolution of just over five minutes (five sets of one sample and one filter measurement scans). The response factor (RF) of NO$_3^-$ and relative ionization efficiencies (RIE) of NH$_4^+$, and SO$_4^{2-}$
were determined following standard operating procedures (COLOSSAL) for ammonium nitrate and ammonium sulfate calibration. OA RIE was experimentally determined through comparison with another PM$_1$ Q-ACSM combined with use of a state-of-the-art organic RIE calibration with organic alcohols recommended by the Q-ACSM manufacturer. An RF of 2.81$\times$10$^{-11}$, NH$_4^+$ RIE of 4.15, SO$_4^{2-}$ RIE of 0.61, and organic RIE of 1.9 (default is 1.4) was applied after validation during data ratification in the standard Q-ACSM data analysis process. Composition dependent collection efficiency (CDCE) was
applied following the (Q-ACSM modified) methods of Middlebrook et al. (2012). The uncertainty in the mass concentration

of the non-refractory species is considered ±30%. The 30-min average limits of detection for the Q-ACSM were calculated to be 0.110 µg m$^{-3}$ for NO$_3^-$, 0.175 µg m$^{-3}$ for SO$_4^{2-}$, 0.662 µg m$^{-3}$ for NH$_4^+$, 0.561 µg m$^{-3}$ for OA, and 0.105 µg m$^{-3}$ for Cl$^-$, following the methods of Ng et al. (2011).

### 2.2.3 Aethalometer AE33

The dual-spot aethalometer (Model AE33, Magee Scientific) operates seven different wavelength channels (370, 470, 520, 590, 660, 880, and 950 nm) to provide optical absorption coefficients by measuring light attenuation every minute through a filter tape that has collected aerosol at a flow rate of 5 (±0.4) LPM. The AE33 Dual Spot$^{TM}$ measurement technique allows for the correction of filter loading effects by aerosol in real-time (Drinovec et al., 2015). The 880 nm wavelength channel is classically used to measure light absorbing equivalent black carbon (eBC) (Petzold et al., 2013; Bond et al., 2013), using the

standard mass-specific absorption cross section (MAC) of 7.77 m$^2$/g (Magee Scientific Inc. (2018); Drinovec et al., 2015). Multiple scattering effects of the collection tape are accounted for with the correction value (*C*) of 1.57 that is based on experimental investigations into TFE-coated glass fibre filter tape material (part no. 8050) (Drinovec et al., 2015). The rolling 15-min average was calculated from the 1-min data to reduce noise. This rolling average was used to interpolate eBC concentrations that matched Q-ACSM data points in time.

### 2.2.4 Xact 625


The Xact 625 (Xact, from this point onward) can measure up to 24 elements between silicon and uranium at hourly time resolution and has been evaluated and described in previous studies (Furger et al. (2017); Tremper et al. (2018)). The instrument has a flow rate of 1 m$^3$ h$^{-1}$; the inlet tube is heated to 45 °C when the ambient relative humidity (RH) exceeds 45%, which was usually the case. The samples are collected onto Teflon tape and subsequently analysed using energy

dispersive X-ray fluorescence (EDXRF). The X-ray source used is a Rhodium anode (50 kV, 50 Watt) and the X-ray fluorescence is measured using a silicon drift detector. In this study, the instrument measured the elements As, Ba, Ca, Cd, Ce, Cl, Cr, Cu, Fe, K, Mn, Mo, Ni, Pb, Pt, S, Sb, Se, Si, Sr, Ti, V and Zn in PM$_{2.5}$. Daily automated quality assurance checks were performed at midnight. Further quality assurance checks, such as flow checks and external calibration checks were performed at the start and end of the campaign.

### 2.2.5 Gas analyzers


Nitrogen oxides (NO$_x$) and sulfur dioxide (SO$_2$) were measured throughout the campaign using automated gas analyzers. The NO$_x$ is measured by the *Teledyne Instruments* Chemiluminescent NO/ NO$_2$/ NO$_x$ Analyzer Model 200A which measured NO and NO$_x$ and by calculation NO$_2$ at 5-min time resolution. The total NO (NO$_x$) can be measured and are taken as parts per billion (ppb), and NO$_x$ is converted to µg m$^{-3}$ as NO$_2$ ppb*1.9125 = NO$_2$ µg m$^{-3}$, and NO ppb*1.28 = NO µg m$^{-3}$ (20°C, 1

atm). The SO$_2$ was measured at 1-min time resolution by a *Teledyne API* Model T100 UV Fluorescence SO$_2$ Analyzer that was used throughout the PortAIR project. A small drift in the SO$_2$ baseline was observed over the yearlong campaign, so the

measurements were subsequently corrected using a polynomial function for baseline drift derived from laboratory tests conducted at the end of the campaign.

### 2.2.6 SMPS

The scanning mobility particle sizer (SMPS) characterises the number-size distribution of the ambient aerosol particles. Particles passing through the system are charge neutralized (Fuchs, 1963) (electrical ionizer model 1090, *MSP*) and then sized by electrical mobility through a differential mobility analyser (DMA, *TSI Inc.* model 3080) and finally counted by a condensation particle counter (CPC, *TSI Inc.* model 3010). The SMPS was operated by passing sample air through a multi-tube Nafion® membrane and into the DMA at a sample flow of ~ 1 LPM with a sheath flow of 5 LPM (Collins et al., 2004).

The SMPS was operated at 3-5 min scan duration with *TSI Inc.* AIM software (release version 9.0.0.0) with charge correction applied.

### 2.3 Source apportionment

Positive matrix factorization (PMF; (Paatero, 1997)) was used to apportion the organic aerosol (OA) measured by the Q-ACSM into different emission source categories. The PMF was conducted on the original 5-min time resolution data using

the multilinear-engine (ME-2; (Paatero, 1999)) implemented in the software SoFi (version 9.4.10) (Canonaco et al., 2013). PMF can be expressed by the bilinear factor model (Paatero and Tapper, 1994):

$$X_{ij} = G_{ik}F_{kj} + E_{ij}$$

Where for Q-ACSM data, $X$ is the measured mass spectrum over time (including negative and zero values) with dimensions $i \times j$, $G$ is the time series of non-negative factors ($i \times k$; $k$ is the number of factors), $F$ is the non-negative factor profiles ($k \times j$),

and $E$ is the residuals of the model with the same dimensions as $X$. The least squared algorithm was employed to minimize the value of Q (sum of squared residuals weighted by respective uncertainties), ensuring a good fit between the model and observed data (Canonaco et al., 2013; Crippa et al., 2014).

In this study, unconstrained PMF solutions were first considered (see Supplementary Fig. S2), but did not yield any

physically meaningful separation of factors. Reference mass spectral profiles were used to constrain the ME-2 algorithm (Canonaco et al., 2013) and these reference profiles were left to vary within specified limits using the limits approach (Lin et al., 2021). Different from the a-value approach where all *m/z* in the mass spectrum vary uniformly, in the limits approach, each *m/z* in the input mass spectrum was individually varied. For example, one *m/z* may have a variation of 2% while another may vary by 40%. This approach is commonly used to capture variation in emission conditions, such as different

stove type for burning solid fuels, and can be found when combining multiple profiles into a mean mass spectrum with standard deviations ($\sigma_j$) at each ion ($m/z_j$). The limits were then set for each *m/z*, with the lower limit ($\overline{m/z_j} - \sigma_j$) and upper limit ($\overline{m/z_j} + \sigma_j$). To assess the robustness of the PMF solution, a Bootstrap resampling strategy was employed

(Paatero et al., 2014; Ulbrich et al., 2009; Davison and Hinkley, 1997; Efron, 1979). This method evaluated the statistical uncertainty of the solution, which could, e.g., arise from variations in emission sources.

For the PMF analysis, an inorganic and organic combined matrix was employed, which combined OA ions and directly measured fragment ions for $SO_4^{2-}$ for an organic-sulfate input matrix. The OA mass spectrum was extended up to $m/z$ 120 and additional columns were added for $SO_4^{2-}$. The error matrix for these ions was generated using the same initial error calculation as for OA. The organic-sulfate input was down weighted cell-wise based on the signal to noise ratio (SNR), where bad or weak signals with SNR<0.2 (negative and zero included) or SNR<2, respectively, are down weighted by being given proportionately higher error values in SoFi Pro. Overall, the calculated SNR for the sulfate ions shows that $m/z$ 81 for $HSO_3^+$ and $m/z$ 98 for $H_2SO_4^+$ are weak (Fig. S3). Additionally, the $CO_2$ related OA $m/z$ 16, 17, and 18 were removed to run the PMF but were added back in later using known fragmentation patterns (Chen et al., 2022; Canonaco et al., 2021; Parworth et al., 2015). The $SO_4^{2-}$ fragment ions included were $m/z$ 48 for $SO^+$, $m/z$ 64 for $SO_2^+$, $m/z$ 80 for $SO_3^+$, $m/z$ 81 for $HSO_3^+$, and $m/z$ 98 for $H_2SO_4^+$ (Sun et al., 2012). Since these ions only account for about 54% of the measured $SO_4^{2-}$ (Fig. S4a), the remaining $SO_4^{2-}$ was added back in later to the factors containing $SO_4^{2-}$. The remaining ion fragments for $SO_4^{2-}$ were calculated based on the ion ratio to $m/z$ 80. This ratio was chosen as this is the $m/z$ value with non-weak SNR that shows the most variation between neutralised and acidic $SO_4^{2-}$ (Chen et al., 2019) and that varied over the intensive campaign between neutralised $SO_4^{2-}$ regional episodes and the acidic $SO_4^{2-}$ in plumes (discussed further in the supplementary material). The organic-sulfate input was well captured by the PMF solution, with a slope between factor mass concentration and input of 1.03 (Fig. S4b).

## 3 Results and discussion

Wind direction and speed data obtained during the intensive campaign indicate predominantly West-South-Westerly winds, with several periods also advecting across the two closest ferry berths (Fig. S5). An overview of the high time resolution air quality data from the intensive campaign is shown in Fig. 1. Many high pollution events of short duration were observed, with the peak $PM_1$ mass concentration reaching 252 µg m$^{-3}$. The pollution events typically lasted 5-35 minutes and were driven by OA, often in combination with $SO_4^{2-}$ and other inorganic species. Elemental sulfur (S), vanadium (V), and nickel (Ni) were also present during pollution plumes that contained $SO_4^{2-}$. While the V/Ni ratio was often in the range 2.5-4.0 (Fig. 1), consistent with HFO emissions (Viana et al., 2009; Pandolfi et al., 2011), an appreciable number of pollution spikes occurred when the V/Ni ratio was less than 2.5, suggesting they are not attributable to HFO emissions. The spikes in $PM_1$ occurred in conjunction with increased $SO_2$ and $NO_x$ concentrations, and enhanced aerosol number concentration ($d_p$ =10-500 nm). However, enhanced number concentration did not always result in very high mass concentrations of the aerosol as they were driven by smaller diameter aerosol particles (e.g. Fig. S6). The very local nature of these pollution spikes is verified by comparing the results with those obtained at the urban background site (UCD) where a $PM_1$ Q-ACSM and AE33 were deployed. The comparison (Fig. S7) shows that while most regional pollution events occur simultaneously at both sites,

Dublin port also has unique and localised pollution spikes that do not occur at the UCD site. Thus, source apportionment was used to explore and identify the origins of these short-lived pollution episodes.

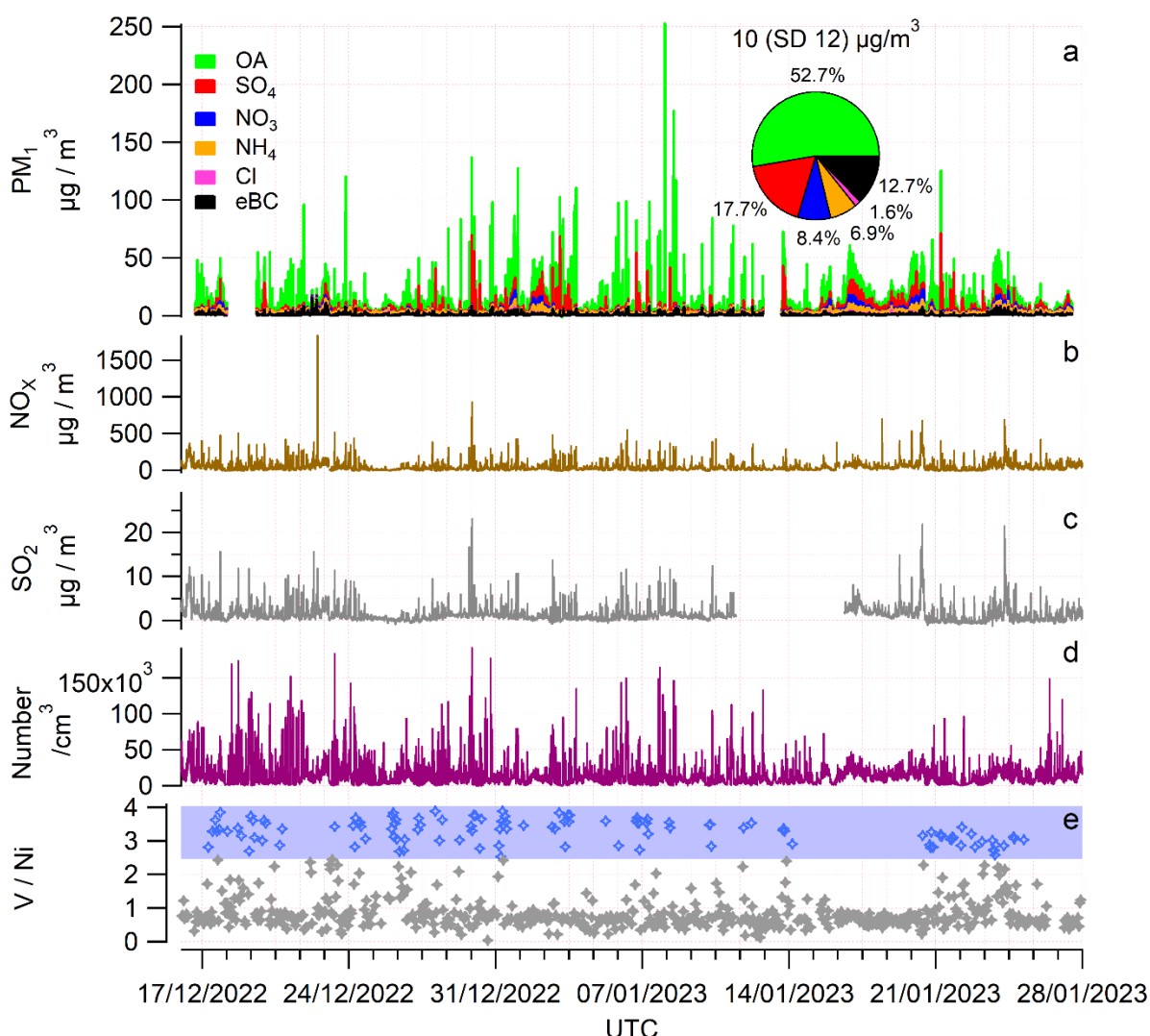

**Figure 1: Time series of the high-time resolution ambient measurements during the intensive PortAIR campaign. From top to bottom panels: (a) reconstructed PM$_1$ on the left axis formed from stacked species along with a pie chart of the average contributions of each (above: mean PM$_1$ (standard deviation)); (b) NO$_x$ in brown; (c) SO$_2$ in grey; (d) number concentration of particles from 10-500 nm in dry electrical mobility diameter (d$_m$) from the SMPS; (e) Vanadium to Nickel ratio (V/Ni) as measured by the Xact with blue shaded area denoting the range of V/Ni traditionally associated with HFO and blue markers showing data in the range (grey when not).**

## 3.1 Ship profile identification

To derive a ship emission profile, data time series were used to search for plumes with known markers including V/Ni ratios, $SO_2$ and $NO_x$ concurrent spikes, and OA with a mass spectral profile indicative of oil or petrol fuel burning. Since the Xact was measuring at hourly time resolution, the V and Ni data were treated as an indicator of a shipping emission plume within that hour. The presence of concurrent spikes in the higher time resolution $SO_2$ and $NO_x$ data was subsequently used to determine the time and duration of likely shipping plumes. Using the aforementioned markers, around 50 plumes were manually identified with a V/Ni ratio in the expected range for HFO emissions and occurred when the wind direction was primarily from the South (Southwest to Southeast included), inferring advection of plumes from nearby ferry berths, the marine shipping channel, and South Dublin Port. However, there were many OA-dominated plumes that lacked V and Ni in either significant concentrations or when the ratio was lower than the expected range for HFO. In these cases, the OA-dominated plumes still contained concurrent spikes in $SO_2$ and $NO_x$ concentrations, and occurred when the wind direction came from the South-Western side of the port across a nearby ferry berth or at times when ships were either in the process of docking or docked. Since the classical V/Ni ratio may no longer be a reliable marker for emissions from ships using low sulfur marine fuels (Anders et al., 2023; Czech et al., 2017), the results obtained here were used to categorise two different types of ship plumes as follows:

**S-Ship** – Sulfate-rich Ship emissions that are characterised by elevated V (0.55 - 0.17 µg m$^{-3}$) and Ni (0.16 - 0.05 µg m$^{-3}$) concentrations, have the well documented V/Ni ratio of 2.5–4.0 associated with HFO, have high elemental sulfur concentrations, and have elevated $SO_2$ and $NO_x$ concentrations. These pollution spikes are also associated with significant concentrations of particulate $SO_4^{2-}$ relative to OA.

**O-Ship** – Organic-rich Ship emissions that are dominated by OA, have elevated $SO_2$ and $NO_x$ concentrations, but do not have the V/Ni ratio associated with HFO and with significantly lower V (< 0.04 µg m$^{-3}$) and Ni (< 0.02 µg m$^{-3}$) concentrations.

To derive the Q-ACSM mass spectral signatures for S-Ship and O-Ship, five exemplary plumes of each type were selected for detailed analysis. The strict criteria for selecting the exemplary plumes were; (i) mean PM$_1$ concentration was greater than 20 µg m$^{-3}$, (ii) the Q-ACSM sampled the plume for at least two data points (more than five minutes), (iii) the plume occurred when the two closest ferry berths had ships manoeuvring in and out of docks or docked at port, as confirmed by Dublin Port shipping logs, and the wind direction was from these respective locations. Additionally, the selected plumes had significantly high OA and $SO_2$ concentrations but were isolated plumes without overlapping regional pollution. The characteristics of the exemplary plumes are described in Table S1. The five exemplary S-Ship plumes had an average PM$_1$ concentration of 61 ± 36 µg m$^{-3}$, with the following composition: $SO_4^{2-}$ (52%), OA (41%), eBC (6%), $NO_3^-$ (1%), Cl$^-$ (0.4%)

and near zero $NH_4^+$ contribution, indicating the plumes were acidic. The five exemplary O-Ship plumes had an average $PM_1$ concentration of $114 \pm 29$ µg m$^{-3}$, with the following composition: OA (92.5%), eBC (6%), $SO_4^{2-}$ (2%), $NO_3^-$ (0.4%), Cl$^-$

(0.2%), and $NH_4^+$ (0.2%). These O-Ship plumes were also acidic with extremely low $NH_4^+$ contribution. The mean OA profiles for ship O-Ship and S-Ship are compared in Fig. S8a. While the OA mass spectrum (in unit mass resolution (UMR)) was similar ($r^2$=0.688), S-Ship contained more signal intensity at $m/z$ 15, 17, 18, 27, 44. Yet, it was apparent that the S-Ship and O-Ship mass spectral profiles of OA showed low variance from each other, which could make them hard to distinguish by PMF if only the OA ions are used in the model matrix. Since S-Ship emissions also had a very strong $SO_4^{2-}$ contribution,

the combined OA and $SO_4^{2-}$ data was used to derive the final ship profiles (Fig. 2). O-Ship $SO_4^{2-}$ ions were present at low relative contributions, but since the profile did not show any realistic fragmentation pattern ($SO^+$ 8.24× 10$^{-3}$, $SO_2^+$ 4.96 × 10$^{-3}$, $SO_3$ 5.23 × 10$^{-3}$, $HSO_3^+$ 1.505 × 10$^{-2}$, and $H_2SO_4^+$ -9.96 × 10$^{-4}$) the ion fragments were set to zero with standard deviation shown in Fig. 2. A comparison was made between the S-Ship and O-Ship mass spectral profiles obtained in this work with the ship profile derived from ACSM measurements in Dunkerque, France (Zhang, 2016), which is also in a SECA zone. The

O-Ship profile compared extremely well ($r^2$ = 0.986) to the Ship-like OA (Sh-OA) profile obtained in Dunkerque (Fig. S9) and confirms O-Ship and Sh-OA as a good reference profile for low-S ship fuel emissions.

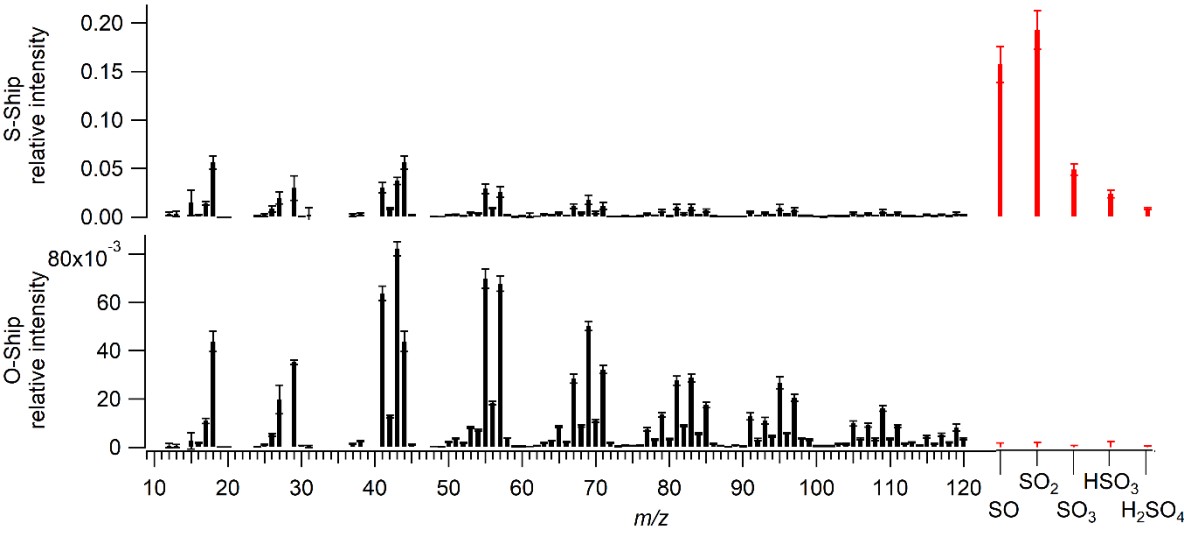

**Figure 2: Reference mass spectra profiles of S-Ship (top) and O-Ship (bottom) plumes. The OA mass spectra are shown in black. The sulfate-related ions $SO^+$, $SO_2^+$, $SO_3^+$, $HSO_4^+$, and $H_2SO_4^+$ ($m/z$ 48, 64, 80, 81, and 98, respectively (see methods in Sun et al. (2012)) are shown in red and placed at $m/z$ 125, 130, 135, 140, and 145, respectively to run PMF. Error bars indicate the standard deviation of the sample from mean for the five exemplary plumes.**

### 3.2 Organic-sulfate source apportionment

**3.2.1 PMF results**

Unconstrained PMF solutions with 2–10 factors were tested as a first step (see Supplementary discussion S1.1). Only the two-factor solution resulted in reasonable profiles, comprised of generic hydrocarbon-like organic aerosol (HOA) and an aged organic aerosol. The rest of the solutions resulted in some particular ions separating out as individual factors that, however, are not physically meaningful. Despite the measurements being conducted in a port environment, no sea salt factor

(see Supplementary) was resolved by the unconstrained PMF solution for the intensive campaign period. This was supported by evaluation of elemental Cl (Xact) that showed comparable contributions from the directions of the city and of the sea.

To direct the PMF model towards a physically meaningful solution, the mass spectra of reference primary OA factors were constrained using the ME-2 algorithm (Canonaco et al., 2013). The organic-sulfate combined PMF was run by combining

the two constrained ship type factors with other constrained factors expected to be present. These include a traffic HOA derived from a previous curb side study in Dublin, Ireland (see Fig. S10) (Lin et al., 2020), as well as individual solid fuel burning (SFB) factors for peat, wood, and coal from a previous Irish study (Lin et al., 2021), a sea-salt factor, and the S-Ship and O-Ship factors discussed above. While finding the most reasonable solution is somewhat subjective, the best solution occurs when increasing the number of factors leads to avoidable splitting of the factors or when reducing the number of

factors leads to avoidable mixing of factors. Whether factors are split or mixed in a solution was evaluated by looking at the solution residuals, correlations to other factors as well as external time series (e.g. $NO_3^-$, $NH_4^+$, eBC, metals, etc.) and checking if diurnal patterns looked representative of real port or city activities (e.g. traffic patterns). The best solution was determined to be with 6-factors, two unconstrained and four constrained factors: S-Ship emissions, O-Ship emissions, Peat, and traffic HOA. Increasing the number of factors for the ME-2 solution beyond six could not resolve any more reasonable

solutions, with extra factors being separated into unrealistic profiles with poor correlation to external tracers. The six-factor solution was then run with bootstrap resampling (50 runs) and found to be very stable, where the standard deviation of the profiles or time series was 2-23%. The factor profiles derived from the 6-factor bootstrap solution for the organic-sulfate source apportionment are shown as factor profiles in Fig. 3a. The time series and diurnal trends for the four constrained factors (S-Ship, O-Ship, HOA, Peat) and the two unconstrained factors (OOA, X-Ship) are presented in Fig. 3b and Fig. 3c,

respectively.

The diurnal variations (Fig. 3c) between the S-Ship and O-Ship factors were similar, while the time series shows differences in the patterns observed for the factors, as well as some periods where the factors overlap but peak at slightly different times. This may indicate intrinsically linked emissions from different emission sources. The HOA factor had a diurnal pattern with

small peaks occurring at the same times as the peaks in the ship factors, which is not surprising given the flow of vehicular traffic linked with ship arrivals and departures. However, the HOA factor also had an evening peak, which could be caused

by the HOA traffic mass spectra being very similar to those for home heating oil at UMR and $m/z < 120$ (Lin et al., 2020). The correlation matrices (Fig. S11) showed HOA correlating with SFB-related factors (peat, OOA) and elemental tracers (As and K), as well as with shipping-related tracers (i.e. $SO_2$ and the X-Ship factor). The HOA factor seemed split between
traffic from the port and a city source that peaks in the evening, likely oil burning for residential heating. Peat showed time trends that match the regional pollution episodes in Fig. S7 and diurnal patterns that are dissimilar to the ship emissions, with a clear evening peak. The increase in the evening is expected for factors associated with residential SFB for home heating.

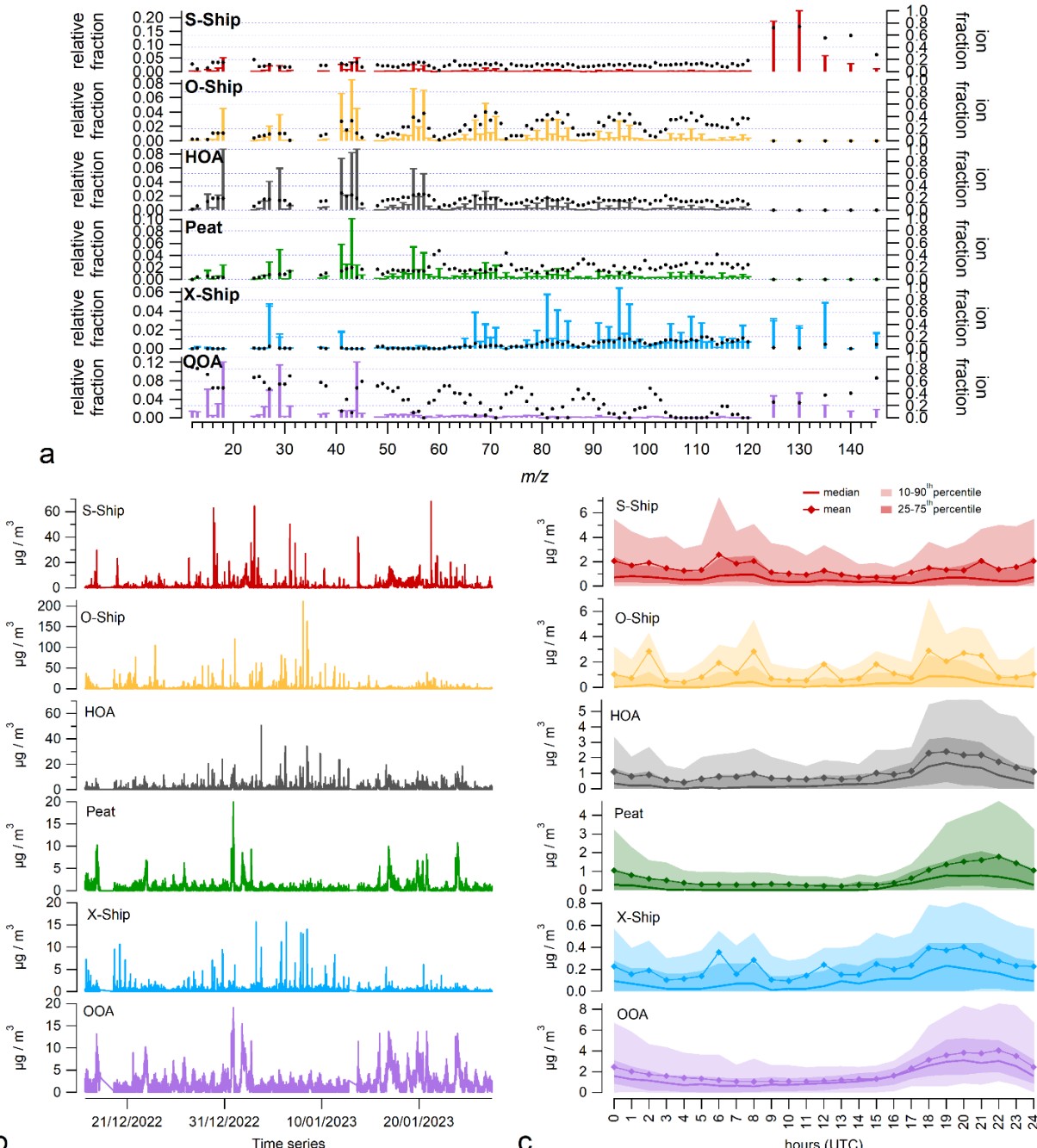


**Figure 3: The 6-factor PMF solution showing (a) factor profiles, (b) the factor time series and (c) the diurnal profiles. The colors of each factor are consistent throughout; S-Ship in red, O-Ship in orange, HOA in charcoal, Peat in green, X-Ship in blue, and OOA in purple. In (a), the left axis is the relative ion fraction of the mass spectrum, with error bars indicating the variation from the bootstrap resampling. Right axis is the relative ion fraction of each *m/z* in that profile compared with the total for that *m/z***

**(markers). In (b), the left axis shows mass concentration ($\mu$g m$^{-3}$) as a function of time for each factor. In (c), the left axis shows the median and mean diurnal cycle of the full time series with 25-75th percentiles (dark shaded) and 10-90th percentiles (light shaded) indicated.**

One of the unconstrained factors is attributed to oxygenated OA (OOA). This factor followed a very similar diurnal pattern and time series to Peat, which points to OOA being present or produced during SFB pollution episodes. The factor correlation matrix (Fig. S11) also supports this assertion, with strong correlations for OOA with Peat (r = 0.74), elemental As (r = 0.64), elemental K (r = 0.62), and Cl$^-$ (r = 0.62) potentially related to the build-up of local pollution, which can be dominated in winter by SFB related pollution (e.g. Lin et al. (2023)). On the other hand, the correlation of OOA with NO$_3^-$, NH$_4^+$ and Cl$^-$ could be related to regional pollution during stagnant weather, so all or some portion of OOA may be independent of SFB. The strong correlation of OOA with NO$_3^-$ (r = 0.77) could indicate the OOA was semi-volatile, freshly formed secondary OA, which would point to the formation of OOA along the route to the port. As a result, OOA is probably a combination of contributions from regional secondary production, including from SFB sources, and freshly formed secondary aerosol.

The second unconstrained factor contained heavier ions, without fragmentation at lower $m/z$, and lacked $m/z$ 44, indicating no ageing. This factor time series is well correlated with O-Ship and to a lesser extent with HOA and S-Ship. It is unlikely to be a split factor due to association with both shipping factors, which points to a source related to ship engines, mostly to vessels using low sulfur fuels. For these reasons, we call this factor X-Ship. The ions at $m/z$ 81 and 95 are typical for exo-sulfur aromatics while there are also $C_nH_{2n-1}^+$ ions for $m/z$ 41 and 55, as well as $m/z$ 105 and 119 that could be carboxylic acids, possibly naphthenic acid. Some of these ions point towards this factor being an indicator of engine oil lubricant (Anders et al., 2023; Czech et al., 2017). The X-Ship OA ions are poorly correlated with the S-Ship ($r^2$ = 0.007) and O-Ship ($r^2$ = 0.089) factors (see Fig. S8b and Fig. S8c). In both cases, the majority of X-Ship ions, especially at higher $m/z$, have stronger relative intensities. The major difference is that X-Ship does not contain $m/z$ 18, 41, 43, 44, 55, and 57, indicating a lack of hydrocarbon content and ageing. The X-Ship factor also appeared in unconstrained PMF runs of the matrices, therefore, it was mathematically divergent and found in most solutions with a few unconstrained factors.

### 3.2.2 Source apportioned ship plumes

The organic-sulfate PMF resolved 58 S-Ship plumes and 190 O-Ship plumes over the intensive campaign. The average chemical breakdown of PM$_1$, along with NO$_x$ and SO$_2$, is shown in Fig. 4a for S-Ship and Fig. 4b for O-Ship. The S-Ship factor plumes were comprised mostly of SO$_4^{2-}$ (57%), followed by OA (35%), eBC (6%), NO$_3^-$ (1%), with negligible contributions from NH$_4^+$ and Cl$^-$. There is slightly more SO$_4^{2-}$ and less OA than the average of the exemplary S-ship plumes, caused by the inclusion of plumes with lower PM$_1$ concentrations. O-Ship plumes were comprised mostly of OA (77%), followed by eBC (9%), SO$_4^{2-}$ (7%), NO$_3^-$ (3%), NH$_4^+$ (3%) and Cl$^-$(1%). The increased contribution from inorganic species compared to the average of the exemplary O-Ship plumes is caused by the presence of plumes on top of regional secondary aerosol. Overall, 7% of the S-Ship plumes and 27% of O-Ship plumes had PM$_1$ concentrations less than 15 µg m$^{-3}$. The 99$^{th}$ percentile of PM$_1$ statistically represents extreme pollution episodes, which was PM$_1$ > 53.5 µg m$^{-3}$ during the PortAIR intensive campaign. The O-Ship factor included 33 plumes where PM$_1$ reached at least 53.5 µg m$^{-3}$ (99$^{th}$ percentile of PM$_1$). Whereas the S-Ship factor only had 10 plumes where PM$_1$ reached 53.5 µg m$^{-3}$.

Using the PMF solution to identify S-Ship and O-Ship plumes, particle mass concentration and other components of these two Ship factor types can also be compared. In the PMF solution, despite the higher frequency of occurrences of O-Ship pollution plumes, S-Ship and O-Ship plumes had nominal average $PM_1$ concentrations of $29 \pm 22$ µg m$^{-3}$and $32 \pm 26$ µg m$^{-3}$, respectively. This was due to S-Ship pollution events having significant fractions of $SO_4^{2-}$, such that mean OA was $10 \pm 10$ µg m$^{-3}$ and mean $SO_4^{2-}$ was $17 \pm 12$ µg m$^{-3}$ for S-Ship. O-Ship however had $25 \pm 24$ µg m$^{-3}$ of OA and $2 \pm 2$ µg m$^{-3}$ of $SO_4^{2-}$ on average. S-Ship $PM_1$ ranged from 9-135 µg m$^{-3}$ and O-Ship ranged from 4–252 µg m$^{-3}$.

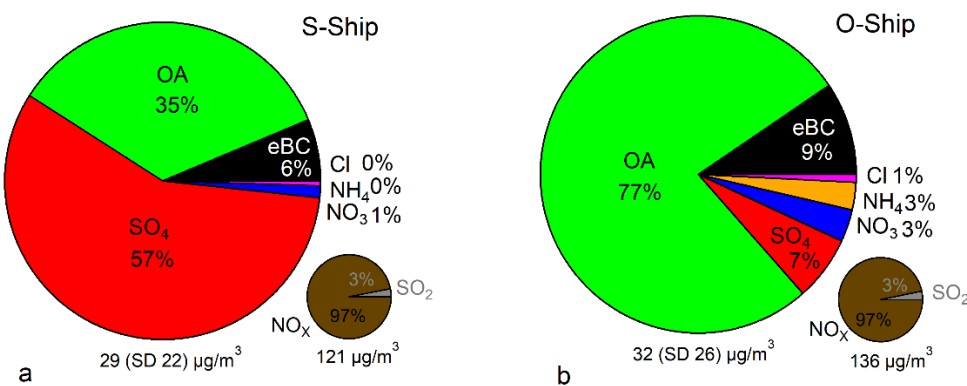

Figure 4: Pie charts of the composition breakdown of $PM_1$ for (a) S-Ship plumes and (b) O-Ship plumes with gas data pie of $SO_2$ and $NO_x$ also shown, respectively.

Average particle number concentrations were $(1.52 \pm 1.55) \times 10^4$ cm$^{-3}$ and $(3.75 \pm 2.28) \times 10^4$ cm$^{-3}$ for S-Ship and O-Ship respectively. Although the number concentration varied widely across the ship plumes, on average O-Ship had more particles associated with this plume type. Additionally, while there were variations in size modal distributions over the duration of the plume and larger variation from plume to plume, the number-size distributions of O-Ship emissions were shifted to smaller sizes than S-Ship (Fig. 5). The combined smaller diameter particles at higher number concentration from O-Ship could have more adverse health impacts as these particles ($d_m < 100$ nm) penetrate into the bloodstream and translocate to all organs in the body (Schraufnagel, 2020). While it seems that some variation in number-size distribution and absolute number concentration may be due to fuel type, it is important to note that additional factors such as variations in engine loadings and the use of lubricating oil may also influence the emissions patterns. Furthermore, the measurements in this study are expected to represent diluted and mixed plumes after undergoing transport, since these are ambient plume detections rather than measurements directly from the stack.

In terms of $NO_x$ and $SO_2$, both median and mean values were similar for the two types, with S-Ship plumes having an average concentration of $(117.6 \pm 118.0)$ µg m⁻³ of $NO_x$ and $(3.5 \pm 3.4)$ µg m⁻³ of $SO_2$ and O-Ship having an average of $(132.3 \pm 108.7)$ µg m⁻³ of $NO_x$ and $(3.9 \pm 3.2)$ µg m⁻³ of $SO_2$. The eBC concentrations were also similar with S-Ship having $(1.8 \pm 1.4)$ µg m⁻³ and O-Ship having $(3.0 \pm 1.8)$ µg m⁻³ of eBC on average. The V/Ni median ratio for S-Ship events was 3.41 (range 2.7–3.9) in line with the literature, but was 0.74 (range 0–2.3) for O-Ship, which is in line with a study that found

V/Ni = 0.6–1.1 after the Global Sulfur Cap 2020 regulation (Tauchi et al., 2022). A summary of the mean characteristics of the two types of ship plumes can be found in Table S2.

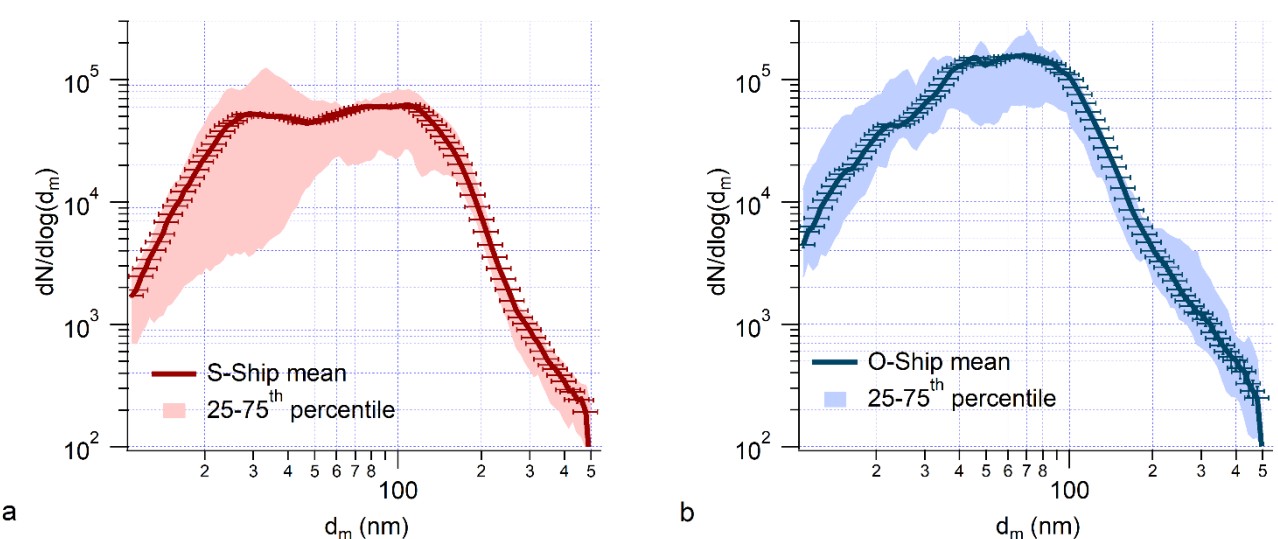

**Figure 5. Number-size distributions showing mean (error bars show uncertainty) and 25th-75th percentiles of the exemplary (Table**
**S1) pollution plumes for (a) S-Ship and (b) O-Ship emissions. The y-axis is average sum of the number concentration in log space ($dN/dlog(d_m)$), and the x-axis is the particle dry electrical mobility diameter ($d_m$) in nm. Percentiles are noisy due to low sample size (n=19 (S-Ship) and n=15 (O-Ship)).**

### 3.2.3 Quantification of sources

Factors from the PMF, inorganic species, and eBC were used to quantify the relative contribution of sources to measured $PM_1$ (Fig. 6). Since the PMF only included the directly measured $SO_4^{2-}$ ions, the $SO_4^{2-}$ fragments that are normally calculated from the measured ones were added back in after to get the true $SO_4^{2-}$ mass concentration. The S-Ship factor and OOA factor contained all significant fractions of the $SO_4^{2-}$ ions (Fig. 3a), so these factors were affected by the re-addition of the non-measured $SO_4^{2-}$ fragments (see Sect. 2.3 and supplementary discussion). The $PM_1$ breakdown using these factors from the

PMF is shown in Fig. 6a.

However, the factors from the PMF are not true representations of the mass contribution of sources since they only take into account OA-$SO_4^{2-}$ input. Therefore, we use the time series of factors from the PMF and other ambient observations to quantify the relative contribution of sources, separating them into primary and regional sources where possible. As outlined in detail below, the procedure for quantification of the sources included separating regional $SO_4^{2-}$ from S-Ship $SO_4^{2-}$ and

included attributing a fraction of eBC to the ship emissions. The result of this procedure is the source contribution estimates shown in Figure 6b.

The PMF solution has trouble separating the regional $SO_4^{2-}$ from S-Ship $SO_4^{2-}$. This is partly because the measured $SO_4^{2-}$ ions have low variability in the ACSM mass spectra, with ammonium sulfate and acidic sulfate having similar relative

intensities (Chen et al., 2019) and fragmentation patterns (see discussion in supplementary material and Fig. S12). The overall effect on the data is that the relative fraction of S-Ship was overestimated by about 8% of the total owing to regional $SO_4^{2-}$. This includes various nighttime periods of elevated regional sulfate that were not attributed to solid fuel burning because coal is the only solid fuel used in Dublin that produces appreciable amounts of sulfate (Trubetskaya et al., 2021) and it was found to have a negligible contribution to $PM_1$ during PMF runs.


While the separation of regional and ship sulfate is a limitation of the organic-sulfate PMF, we can still isolate the real S-Ship emissions by manually subtracting the contribution of regional $SO_4^{2-}$ from the S-Ship factor. This adjusted S-Ship is called S-Ship* and reflects S-Ship emissions with regional $SO_4^{2-}$ subtracted out. The pollution rose of S-Ship* indicates an emission source from a nearby ferry terminal (Fig. S13 and S14), and no longer includes a nighttime increase in the diurnal

profile (Fig. S14).

Since eBC is observed to be part of the ship emissions, it was apportioned to the S-Ship and O-Ship factors using the estimated eBC/(OA+$SO_4^{2-}$) ratios (eBCr). The ratios from the observed five exemplary plumes for S-Ship (eBCr = 0.066) and O-Ship (eBCr = 0.047) were compared to those derived from the source apportioned S-Ship and O-Ship plumes. The S-

Ship was comparable with an average eBCr of 0.068, close to 0.066 from the S-Ship five exemplary plumes. However, the O-Ship eBCr differed from 0.047 for the five exemplary plumes to an average of 0.113 for the source apportioned plumes. This difference was caused by the much larger OA contribution during the exemplary plumes than for the average taken over all O-Ship factor plumes. Therefore, one ratio was applied for all O-Ship plumes where $PM_1 > 53.5$ µg m$^{-3}$, and for the rest of the period another ratio was applied, eBCr = 0.063 and eBCr = 0.141, respectively. These eBCr ratios were used to re-

apportion eBC, amounting to 11% of total eBC to S-Ship* and 13% to O-Ship*, which denotes O-Ship plus the eBC fraction. The eBC shown in the pie chart (Fig. 6b) is adjusted (eBC*) and reflects a subtraction of the 24% of eBC (only 2.9% of $PM_1$) already accounted for in the ship fuels. Nevertheless, the pollution rose of eBC* (Fig. S13) suggests a significant influence from the East-Southeast that could reflect shipping related traffic through Dublin Port and the wider channel.

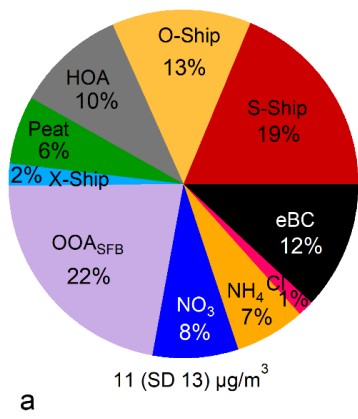 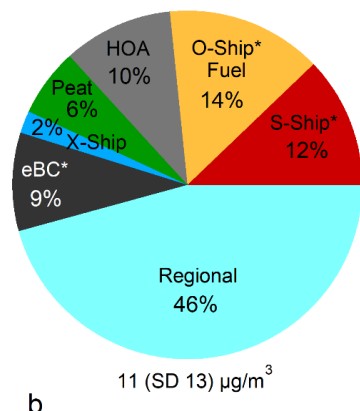


**Figure 6: Pie chart of the composition breakdown of (a) the PMF factors and other species contributing to PM₁ and (b) source contribution estimates to PM₁ with adjusted O-Ship, S-Ship, and eBC. Mean PM₁ is shown at the bottom of the pie (brackets show standard deviation of the mean).**


Factors and chemical components that belong to regional sources were combined and called Regional source, which includes the OOA, $NO_3^-$, $NH_4^+$, $Cl^-$, and regional sulfate subtracted from S-Ship. Just to note, an estimated 1% of PM₁ from the combined $NO_3^-$, $NH_4^+$, and $Cl^-$ could be associated with O-Ship fuel emissions rather than Regional, however, there was enough uncertainty in this estimation that the 1% has been left associated with Regional. The pollution rose of the regional

source shows two major source origins, one from the west and one from the northeast, and very little from the ocean or nearby ferry terminals, which also confirms that most of the inorganics (other than sulfate) did not originate from shipping emissions. The S-Ship* and O-Ship* contains the apportioned eBC, as explained above. Across the campaign, the source contributions to the measured PM₁ were Regional (46%), O-Ship* emissions (14%), S-Ship* emissions (12%), HOA due to traffic or oil burning (10%), eBC* (9%), Peat (6%), and X-Ship emissions (2%). The Dublin Port ship-related factors made up

28% (S-Ship*, O-Ship*, and X-Ship) of PM₁, not counting ship traffic related HOA and associated port traffic and shipping lane eBC. It was difficult to attribute the HOA factor in Dublin Port to either ship-related traffic, city traffic, or oil burning for residential heating, as the pollution roses only indicate a local and often westerly source (Fig. S13). Therefore, we estimate that shipping-related emissions in Dublin Port contributed 28–47% of PM₁ (Fig. 6b), where the upper range represents the entire contribution of hydrocarbon-like organic aerosol (HOA) and traffic-related eBC added to the estimate

(from activities such as ferry traffic, vehicles for moving containers, and crane engines).

### 3.3 Ship emissions when manoeuvring and hoteling

With the unique methodology in this study, PMF was able to identify and separate O-Ship and S-Ship plumes, which allowed the frequency of the different ship emissions to be evaluated. Furthermore, by using the shipping logs from Dublin

Port Company along with information on fuel types used by specific ships, the emissions from individual vessels were

isolated and investigated for steady and favourable wind conditions. Using this approach, O-Ship emissions were attributed to low-S fuels, mainly VLSFO, and S-Ship emissions were attributed to HFO with wet scrubber devices.

In Dublin Port, ship manoeuvring takes on average $30 \pm 10$ minutes from the outer buoy to docking. Additionally, there is no shore power or 'electric ironing' in Dublin Port, so the ships run their engines or generators when at dock, a process called

'hoteling'. Ships will enter the port at reduced speeds from cruising in the open ocean. Previous studies have shown that $PM_1$, $NO_x$, and black carbon all decrease with decreasing ship speed, while $SO_2$ and particle number remain constant over speed (Cappa et al. 2014; Wu et al. 2021). So, the emissions from ships are expected to be different during manoeuvring because the engine is under a different load compared to hoteling or cruising (Anderson et al., 2015; Cappa et al. 2014; Liu et al., 2018). Additionally, since ships in Dublin port are hoteling in a SECA it is common for ships to switch to different

engines with compliant fuels after docking (typically MGO in Dublin Port). Thus, it was interesting to note that, when meteorological conditions were steady, the manoeuvring in and out of the nearby berths showed distinct plumes with idling periods in between, resulting in time series of both mass and number concentration that resembled the shape of 'bat ears' (Fig. 7). The bat ear profiles are characterised by intense plumes of $PM_1$, $NO_x$ and $SO_2$ during inbound and outbound manoeuvring with a large drop in concentration in between where $PM_1$ fell typically below 15 µg m$^{-3}$ and the gas

concentrations also dropped but remained elevated above background. This drop in $PM_1$ is partially to do with the sampling technique used by the ACSM, as it has a lower particle size cut-off of roughly 40 nm vacuum aerodynamic diameter (equivalent to ~ 32 nm electrical mobility diameter ($d_m$) (DeCarlo et al., 2004)).

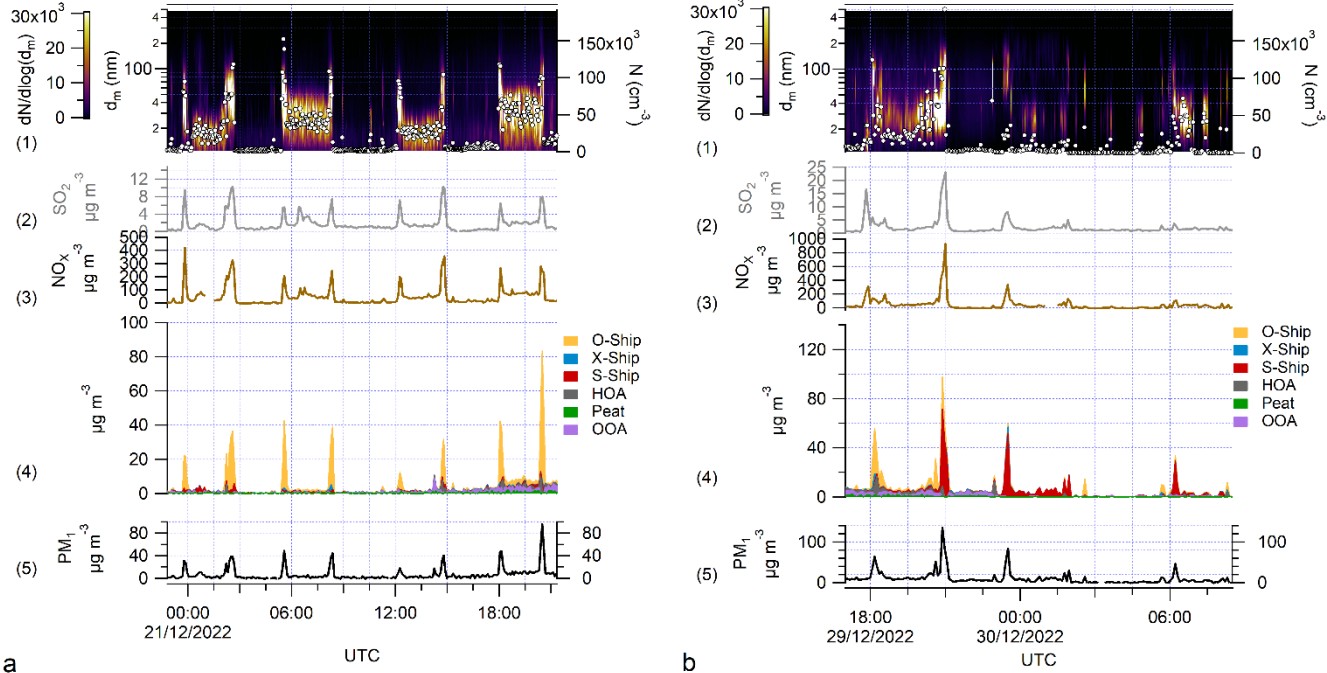

**Figure 7: Time series of data during 'bat ear' ship emission events for (a) isolated O-Ship plumes (fuel switching pattern VLSFO-MGO-VLSFO) and (b) both O-Ship and S-Ship plumes. Panel (1) shows a curtain plot of particle number-size distribution data with particle diameter ($d_m$ (nm)) (left axis) and lognormal particle concentration ($dN/dlog(d_m)$) indicated by color. The particle number concentration ($cm^{-3}$) is shown as white circles (right axis). Panel (2) shows $SO_2$ data (grey) and (3) shows $NO_x$ data (brown). Panel (4) shows factors from PMF (as in Fig. 3), and panel (5) shows $PM_1$.**

The O-Ship 'bat ear' profiles in Fig. 7a were the result of VLSFO-powered vessels switching to marine gas oil (MGO) when docked. While not as visible in the ACSM sampled mass concentration, MGO emissions are visible in the number-size distribution as large concentrations (e.g. $6 \times 10^4$ $cm^{-3}$) of tiny particles with $d_m < 50$ nm (Fig. S15). It is also important to note that, these hoteling periods are characterised by particles with $d_m < 32$ nm contributing 51-100% of the estimated SMPS particle volume, which is smaller than the lower size cut-off of the aerodynamic lens in the ACSM. This means that a

significant portion of MGO particle emissions are not captured by the ACSM and do not contribute to the reconstructed $PM_1$ mass concentration or source contribution estimates shown in Figure 6. Nonetheless, estimates of the mass concentration during the four hoteling periods (in chronological order) have been determined from the SMPS data (assuming spherical OA particles with a density of 1.27 g $cm^{-3}$) to be 11, 30, 15, and 54 µg $m^{-3}$ respectively.

The first and third bat ear profile in Figure 7a were from one specific vessel and the second and fourth from another, which may reflect differences in the fuel or engine design. Fewer bat ears attributed to S-Ship emissions were captured due to the lower frequency of these vessels and changing meteorology, but a few are shown in Fig. 7b. At the start of the time series in this period, the wind direction was originally crossing a VLSFO vessel berth before changing to pass over a berth with a HFO-powered vessel and staying steady. The emissions observed from 23:00 on December 29th to 02:00 on 30th December

(Fig. 7b) originated from a vessel that used HFO with a wet scrubber system (closed loop) all the time. This emission profile was created with different engine loads, as fuel type did not change. The preceding (December 29[th] ~21:00) and following (30[th] December ~06:15) S-Ship bat ear emission profile was from a vessel that used HFO (with scrubber) during manoeuvring and MGO for electricity generation when in port. Overall, these patterns confirm that O-Ship emissions from vessels using low-S fuels yield higher concentrations of particles that are shifted to smaller sizes, while S-Ship emissions from HFO-powered vessels with scrubbers often yield relatively less particles (as seen in Fig. 5).

## 4 Conclusions

This work shows that a combination of organic-sulfate PMF is effective in identifying and separating sulfate-rich ship (S-Ship) and organic-rich ship (O-Ship) emissions in a SECA port, and, thus, can help in quantifying their contributions to PM even when S-Ship and O-Ship mass concentrations are as low as 4 µg m$^{-3}$. Over the month-long winter intensive campaign, 58 S-Ship plumes and 190 O-Ship plumes were identified, of which 43 reached over 53.5 µg m$^{-3}$ of PM$_1$ (33 O-Ship and 10 S-Ship). Close investigation of source apportionment factors, information on vessel fuel use, wind direction, and shipping logs indicate that the S-Ship relates to ships that use HFO but have a scrubber system, and O-Ship relates to ships that use low-sulfur (low-S) marine fuels, primarily VLSFO. These two distinct types of ship emission profiles enable organic-sulfate source apportionment with the advantage of distinguishing scrubbed HFO emissions from VLSFO fuel emissions without the use of V/Ni tracers.

Ship plumes were observed to last up to 2.5 hours given steady wind direction and fuel use while a ship was in port. Although, the more extreme pollution peaks (PM$_1$ > 53.5 µg m$^{-3}$) only lasted 5–35 minutes and were specifically caused when ships were manoeuvring in or out of berth. While cold ironing periods at dock were characterised by lower PM emissions, number concentrations remained extremely high for submicron particles (d$_m$ < 50 nm), especially when ships were switched over to Marine Gas Oil (MGO) for power. In fact, MGO emissions were only characterised by these large number-concentrations of small particles, as PM mass concentrations were neither clearly noticeable as plumes (PM$_1$ < 15 µg m$^{-3}$) nor from the mass spectra of OA (noisy), mostly due to instrumental limitations of measuring these very small particles with an ACSM. This limitation leads to an underestimation of the contribution of low-sulfur fuels to PM$_1$ and highlights the need for monitoring the aerosol number-size distribution in ship emission studies. Overall, shipping-related emissions in Dublin Port contributed at least 28–47% of PM$_1$ measured in the port location, with the caveat of missing significant PM$_1$ contributions from MGO fuelled ships that make up a significant portion of vessel traffic at Dublin Port. There were also several cases of stagnant and cold weather conditions, lasting days at a time, that resulted in the build-up of regional and city pollutants that were found to contribute 46% of PM$_1$. Despite the transient and short-lived nature of shipping emission plumes in Dublin Port, the S-Ship and O-Ship emissions combine to contribute a surprisingly significant fraction of PM$_1$ and submicron particle number concentration in the port area with a potential to increase even further with the planned port activity expansion. With more and more ships investing in low-S and alternative fuels, future studies on air quality in ports

will be needed to better capture and investigate the very high concentrations of small particles ($< 50$ nm diameter) expected from cleaner fuels, including MGO, as there are potentially serious implications for particle transport, toxicity, and climate forcing.

**Code Availability**

Software for PMF analyses are run using Igor Pro® from WaveMetrics® (https://www.wavemetrics.com/) and SoFi Pro from Datalystica (https://datalystica.com/sofi-pro/) that are available for purchase.

**Data availability**

Raw data from the study as well as details on the analyses are available upon request to the contact author(s) Kirsten N.
Fossum (kirstennicole.fossum@universityofgalway.ie) and Jurgita Ovadnevaite (jurgita.ovadnevaite@universityofgalway.ie).

**Supplement link:**

**Author contribution:**

JO, COD, and JW designed the research. KNF, NOS, and SJ, carried out the measurements with technical support from SH,
AT and DC. AT and DG supplied the Xact and SB supplied the Q-ACSM. KNF and CL performed the analysis. KNF wrote the paper with support from all authors who commented on the paper.

**Competing interests:**

The authors declare that they have no conflict of interest.

**Acknowledgements**

This work was funded by the Irish Environmental Protection Agency (EPA) under grant number: 2020-CCRP-LS.6, AC3/AEROSOURCE project, supported by EPA-Ireland and the Department of Environment, Climate and Communications.
The PM$_1$ Q-ACSM used in the intensive campaign was provided by Steigvilė Byčenkienė and Vadimas Dudoitis from the SRI Center for Physical Sciences and Technology, Lithuania.

KNF acknowledges and thanks Doug Worsnop for advice on selection of PMF solution, Teresa Spohn for on-site support during the PortAIR campaign, Eamon McElroy, Ken Rooney, and John Dungan and colleagues in Dublin Port Company for access, ongoing support and port-related information, Stenaline, especially Eamon Fortune, for providing insight, access to facilities, and accommodating the sampling site, and also to representatives from the shipping companies that provided details of ship fuel use.

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
