# Peer review of "Two distinct ship emission profiles for organic-sulfate source apportionment of PM in sulfur emission control areas"

_EGUsphere, 2024_

## Author Comment (AC1)

**Author Comments in response to Referees Comments in ACPD for egusphere-2024-1262**

Author comments (AC) are written after each referee comment (RC) in **bold.**

**RC1:**

This study presents in-situ measurements taken at a port area within SECA. The authors identified the reference mass spectrum for two distinct types of ship plumes. Based on this, they performed a source apportionment analysis to characterise contributions from both shipping and regional residential emissions. The PMF factors developed in this study will inform future aerosol mass spectrometry research, aligning well with the scope of the ACP journal. However, the discussions and the definitions of terms require improvements to enhance clarity.

**AC: We thank the referee for their time in reviewing and thereby improving this manuscript. We appreciate the suggestions to improve clarity and have implemented changes to this end.**

RC1: Major comments

1.  The terms given to the shipping emission types throughout the manuscript are inconsistent. In the PMF section (Section 3.2), the authors identify the reference mass spectra profiles as "O-ship" and "S-ship" (Figure 3). However, these terms change to "VLSFO" and "HFO" in Figure 5 (Section 3.3), with additional discussions on the usage of "MGO" fuel. In the conclusions section, the authors state in Lines 486-487 that the organic-sulfate PMF can identify "HFO" and "VLSFO" emissions. I suggest the authors revise their terminology and provide clear definitions for each term.

**AC: The difference in terminology from Section 3.2 to 3.3 reflects additional data processing of the PMF-derived factors to determine source contributions more accurately. We accept that the mixed use of PMF factors and fuel types could create some confusion and have therefore reviewed the text and made several amendments. We have clarified the terminology in section 3.3 and now use S-Ship\* and O-Ship\* to denote the source contributions resulting from the related PMF factors, which have been adjusted for regional sulfate and black carbon emissions. The terms HFO and VLSFO in Figure 5 panel 4 (Figure 7 in the revised manuscript) have been removed and replaced with the PMF factors that include S-Ship and O-Ship, respectively. We have also separated the pie charts in Figure 4 and moved another pie chart from the Supplement to the main manuscript to help improve understanding. Figure 4 in the revised manuscript now shows the average composition breakdown of the O-Ship and S-Ship plumes, while a new figure (Figure 6) has been created to show the PMF factors and other species contributing to PM$_1$, as well as the source contributions. Finally, we have also reworded the statement in the conclusions section.**

**We believe the terminology has now been successfully revised and clear definitions provided.**

2. Following comment #1, for Section 3.3, Although the statistical increase in NOx and SO2 concentrations indicated the presence of a ship plume during the "bat-ear" period, the particle mass concentration remained low, possibly because the smaller particle sizes were below the detection limit of the ACSM. Is this low mass concentration period also classified as an "O-ship" plume period? If so, could the low mass concentrations during these periods affect the composition breakdown pie chart results in Figure 4, and potentially underestimating the contribution of O-ship to the aerosol mass concentration?

**AC: The referee has hit on an important point. As presented in Figure 5a in the original manuscript (Figure 7a in the revised manuscript), the measured parameters (particle mass, particle number, NOx and SO$_2$) all showed a spike in concentration when VLSFO powered vessels were manoeuvring in and out of the berth. The O-ship factor accounted for almost all the particle mass measured by the ACSM during these periods. However, when the vessels switched to MGO fuel to power the auxiliary engines when hoteling, the mass concentration dropped to near-baseline levels, while the particle number concentration remained elevated. Inspection of the SMPS data during the four hoteling periods (between the ears), reveals very low particle sizes at high number concentration. A significant fraction of the particles was below the lower size cut-off for the aerodynamic lens in the ACSM (around 40 nm vacuum aerodynamic diameter) and therefore not detected. As a result, these particles do not contribute to the reconstructed mass concentration or pie chart results shown in the manuscript. The manuscript has been amended to incorporate this point.**

**Nonetheless, estimates of the mass concentration during the four hoteling periods (in chronological order) have been determined from the SMPS data (assuming spherical particles with an OA composition and a density of 1.27 g/cm$^3$) to be 11, 30, 15, and 54 μg/m$^3$ respectively. The particles below 40 nm aerodynamic diameter (~32 nm electrical mobility diameter (DeCarlo et al., 2004)) make up ~ 100%, 73%, 100%, and 51% of the estimated submicron mass concentration respectively.**

**According to shipping logs, two vessels were responsible for the "bat-ear" profiles during this period; the first vessel (*Vessel A*) was associated with the first and third profiles, while the other vessel (*Vessel B*) accounted for the second and fourth profiles. The results indicate that during the hoteling period, *Vessel A* produced particulate emissions that were too small to be detected using the ACSM, while *Vessel B* emitted particulates that were only partially detected by ACSM and mainly assigned to the O-Ship factor. This observation indicates that fresh particulate emissions from MGO-powered engines may not be easily observed using the ACSM and that the contribution of low sulfur fuel emissions (represented by O-Ship\* in the revised manuscript) to the measured PM$_1$ is underestimated. This limitation is now presented in the revised manuscript.**

3. Page 16, Line 396 – 402, In addition to the fuel types, other factors such as variations in engine loadings and the use of lubricating oil may also influence the absolute number concentrations. Furthermore, since the measurements were not taken directly at the stack, the absolute concentrations may be diluted during plume transport.

**AC: Yes, we agree. We have now included this perspective in the discussion section.**

4. Page 14, Figure 3, Although all PMF factors have distinctive mass spectrum profiles and generally weak correlations over the entire campaign period, S-ship related OA shows a peak during nighttime (from 8 pm to 3 am), which is similar to the trend of peat-related OA. In contrast, the diurnal pattern of O-ship related OA is more irregular which makes sense.

Given that residential solid fuel combustion may also contribute to sulfate emissions (Lin et al., 2021), is the S-ship factor slightly influenced by regional emissions, particularly during the night period?

**AC: Yes, the S-Ship factor is slightly influenced by regional emissions of sulfate, as discussed in lines 418 – 423 in the original manuscript. During PMF analysis, the possible contributions from coal, peat and wood burning were all investigated using well known OA mass spectral profiles (Lin et al. 2017). While peat was found to contribute 6% to the reconstructed PM1 mass concentration, the amounts of OA from coal and wood burning were negligible. While coal and wood are historically present in Dublin residential areas, this finding is consistent with previous wintertime observations in Dublin showing peat as the dominant contributor (Lin et al. 2020). Since coal combustion is the only solid fuel that produces appreciable amounts of sulfate (Trubetskaya et al. 2021), we are confident that local residential solid fuel combustion does not contribute significantly to the S-Ship factor and assign the observed sulfate to regional emissions. Note that there may be a diurnal pattern to the regional sulfate as with the OOA, but not necessarily from solid fuel burning. This is verified when considering the diurnal trend of S-Ship\* (formerly referred to as HFO), which has the regional sulfate contribution removed, and shows a pattern which is more closely related to the shipping activity than the S-Ship factor (see the supplementary Figure, b, bottom panel). To make this point clearer to the reader we have included this new figure in the Supplementary Information (Fig. S14), which shows pollution roses, time series, and diurnal trends for S-Ship and S-Ship\* for comparison purposes.**

[Figure]

**Figure. Graphs show differences in (a) the S-Ship PMF factor (b) the S-Ship\* source contribution estimate obtained by subtracting regional sulfate from S-Ship. (Top) Pollution roses plotted using the statistical mean in the Openair package in R where the color denotes the weighted concentration of the species based on wind direction and speed. (Middle) Time series of the data. (Bottom) Diurnal trends of the data (mass concentration vs hour of the day) with triangle markers and error bars showing mean and standard deviation, and box plots showing medians and percentiles.**

RC1: Minor comments

1. Page 1, Line 40. Should be contributed to "PM1" instead of total "PM".

**AC: Corrected.**

2. Page 3, Line 64 – 86, though wet scrubbers or poor quality low-sulfur fuel may have some disadvantages, previous studies (e.g. Yu et al., 2020) have shown that the IMO 2020 regulation will lead to significant reduction of aerosol emissions. Please add some information about the positive part of the new IMO regulations.

**AC: Thank you for this comment. We agree that the impacts of the IMO regulations should be presented in a balanced way. To this end, we have added a statement on the aims of the IMO regulations and also highlighted the positive impact that the reduction in pollutant emissions will achieve.**

3. Page 4, Line 100- 105. Dublin port is within SECA (regulated fuel sulfur content < 0.1%), and the authors declare that the VLSFO (fuel sulfur content < 0.5%) is used here for the engine. As 0.5% is the regulated level for the open sea shipping lane, is the fuel with fuel sulfur content larger than 0.1% but smaller than 0.1% allowed to be used within SECA?

**AC: We believe the reviewer meant. '…is the fuel with fuel sulfur content larger than 0.1% but smaller than 0.5% allowed to be used within SECA?'**

**In which case, yes, fuels with sulfur content in the range 0.1-0.5% are allowed according to *Article* 7 of the EU Directive 2016/802 (Port Directive, 2016) which says '*Member States shall take all necessary measures to ensure that ships at berth in Union ports do not use marine fuels with a sulphur content exceeding 0,10 % by mass, allowing sufficient time for the crew to complete any necessary fuel-changeover operation as soon as possible after arrival at berth and as late as possible before departure.*'**

**In keeping with these regulations, VLSFO-powered vessels entering Dublin Port are required to stop using VLSFO as soon as possible after arrival at berth and as late as possible before departure. During the hoteling period in the port, electricity is typically provided by auxiliary engines powered by an ultra-low sulfur fuel oil, such as MGO.**

4. Page 8, Section 2.2.3, Please give more details how the AE33 is corrected for the multiple-scattering factor.

**AC: We use a constant multiple scattering correction factor (*C*) based on the filter tape material as described in Drinovec et al. 2015. We used PTFE-glass-fiber tape (Part No. 8050) with an experimentally determined value of *C* = 1.57 (Drinovec et al. 2015). This information has been added to the text.**

5. Page 16, Line 396 – 402, please bring the SMPS results in Figure S10 to the main context to support your discussions about the size modes of ship plumes.

**AC: Done. Fig. S10 is now Fig. 5 in the revised manuscript.**

6. Page 18, Line 457 – 460, The authors could benefit from incorporating discussions of previous studies that characterise emissions from various ship operation modes (Wu et al., 2021).

**AC: Done. In the revised manuscript we briefly discuss previous work on this topic (Wu et al. 2021 and Cappa et al. 2014).**

**RC2:**

The study analyzed ship emissions at Dublin Port using Aerosol Chemical Speciation Monitoring (ACSM) data and Positive Matrix Factorization (PMF) methods. It identified two main types of ship emissions: sulfate-rich S-Ship and organic-rich O-Ship. S-Ship emissions are associated with the use of heavy fuel oil, while O-Ship emissions are attributed to low-sulfur fuel types. The study is innovative and, after making some minor revisions, it is recommended for publication.

**AC: We thank the referee for their time and constructive feedback. We have implemented all necessary changes as suggested.**

1. The author mentions in Section 3.2.1 that Appendix Figure S2 shows the results for 2 to 10 factors, but where is this information in Appendix Figure S2?

**AC: Thank you for pointing out this discrepancy. This statement has been changed as follows; 'Unconstrained PMF solutions with 2–10 factors were tested as a first step (see Supplementary discussion S1.1).'**

2. The core of this article is based on the ratio of vanadium to nickel and sulfate fragments; however, to my knowledge, ship emissions cannot be solely reliant on the ratio of V to Ni. The V/Ni ratio is also not a standard tracer for ship emissions. Especially since 2020, there have been strict regulations on the fuel used by ships in emission control areas, which necessarily affects the usability of the V/Ni ratio. The author needs to provide more direct evidence. Furthermore, the measurement of vanadium and nickel is conducted in PM2.5, whereas ACSM measures PM1. How does the author account for this discrepancy?

**AC: This study uses a V/Ni ratio in the range 2.5-4.0 to aid the identification of emissions from HFO-powered vessels. This is a well-established parameter which has been widely used in previous port studies (e.g. Mazzei et al., 2008; Pandolfi et al., 2011; Viana et al., 2009). As the referee correctly points out, the V/Ni ratio has recently become a less reliable tracer for ship emissions because the updated fuel regulations have resulted in some vessels switching to lower sulfur fuels which contain different amounts of vanadium and nickel. Therefore, the V/Ni ratios are only used as a starting point in identifying ship emissions from HFO. We then combine other tracers to identify ship plumes including $SO_2$, $NO_x$, OA profiles, and finally wind direction and shipping logs. Section 3.1 outlines this process. However, we have reworded the text in places to be clearer.**

**There is no discrepancy in using a plume tracer derived from PM2.5 measurements, because it relates to the same time period as the PM1 data obtained from the ACSM. Since the V and Ni are also used as a ratio, it is more important that this be consistent with the literature, where studies have derived a V/Ni ratio of 2.5-4.0 for HFO emissions in PM2.5.**

**To conclude, the V/Ni ratio was just used to aid the initial identification of the plumes, but was not used in the PMF analysis, which is the core of this article.**

3. The author spends a considerable amount of content introducing the ship scrubber device, which in my view, is unnecessary to elaborate on to such an extent.

**AC: We aim to keep the text concise but informative, so we thank the referee for this suggestion. In the revised manuscript, we have reduced the text describing ship scrubber devices, while also retaining the essential information for readers that are not familiar with their operation.**

4. In the manuscript, the author notes that Dublin experiences poor air quality daily due to the burning of domestic solid fuel for home heating, particularly during the colder months (mainly winter), especially at night. This could have a significant impact on the results of this study. Since another assumption of this study is that sulfate fragments are primarily derived from ship emissions, the author should provide more convincing discussions.

**AC:  Although Dublin does often experience poor air quality in winter due to residential solid fuel burning, the dominant fuel type is peat, which does not produce sulfate (Trubetskaya et al. 2021). Coal does produce sulfate, but the PMF analysis showed that the contribution of coal and wood burning were both negligible. We are therefore confident that local residential solid fuel combustion does not contribute significantly to the S-Ship factor and assign the other sulfate component to regional emissions. This is now reflected in the pollution roses for S-Ship and S-Ship\* shown in Fig. S14, which highlight a nearby ferry berth in the southerly direction as the main source of ship-related sulfate emissions. The other sulfate component originates from the westerly direction along with OOA, and together these species are both attributed to regional source emissions (Fig S13f).**

5. I didn't see any diagrams or introductions regarding wind direction and air mass trajectories, which makes the reader suspect whether the monitoring site is influenced by emissions from nearby factories.

**AC: Thank you for this comment. We have now provided information on wind speed and direction, along with pollution roses in the Supplementary Information (Fig. S5 and Fig. S14). It is possible that the monitoring site was influenced by emissions from nearby factories, however, the PMF analysis did not indicate any strong evidence for this. While the elemental composition data obtained by the Xact instrument may reveal some information on industry sources, an in depth investigation of this data is beyond the scope of the current paper.**

6. In line 234, the author indicates that a considerable number of pollution peaks have a V/Ni ratio below 2.5, suggesting the use of different types of fuels in the port area. Where does this conclusion come from?

**AC: The V/Ni ratio in the range 2.5-4.0 is a well-established parameter that has been widely used in previous port studies to identify and apportion HFO emissions (e.g. Mazzei et al., 2008; Pandolfi et al., 2011; Viana et al., 2009). For plumes with a V/Ni ratio outside of this range, it is reasonable to assume that a different fuel type is responsible and the use of other factors including the ACSM mass spectral profile, particle number concentration, NOx and SO2, as well as shipping logs and wind direction etc., has been used to confirm this.  However, we take your point that this specific statement is not well justified in the**

**text. It has been altered to read, "…an appreciable number of pollution spikes occurred when V/Ni ratio was less than 2.5, suggesting they are not attributable to HFO emissions".**

7. Line 255. "The comparison (Fig. S6) shows that while some regional pollution events occur simultaneously at both sites, the pollution spikes at Dublin Port are unique and localised.", "Using this approach, around 50 plumes were manually identified with the V/Ni ratio in the expected range for HFO emissions, and occurred when the wind direction was primarily from the South (Southwest to Southeast included)" These two sentences need to be clearly explained. Because from Figure S1, I can see that under higher wind speeds, the factory is likely to contribute some pollutants.

**AC: We thank the reviewer for their comment, but it seems that these two sentences have been conflated. In fact, these are two separate statements. The first sentence (line 255 of the original manuscript) refers to comparison of ACSM-derived data obtained at the Dublin Port site and another location in Dublin, UCD, to identify regional pollution episodes that occurred at both sites. This helped confirm that the spikes in PM1 concentration observed at Dublin Port site were indeed very local and related to nearby sources in the area. The second sentence refers to the approach used to identify shipping plumes using multiple tracers. The V/Ni ratio was only used as an initial tracer along with wind direction to reveal 50 plumes that are likely due to HFO emissions.**

**For further clarification of the relationship between wind direction and PM1 sources, we have added a figure in the Supplementary Information (Fig. S14) which contains pollution roses of the various sources identified. The plots clearly show that the source apportioned shipping factors originate from the direction of nearby berths in Dublin Port.**

8. What does the "4ion fraction" on the right coordinate of the O-ship mass spectrum in Figure 3 mean?

**AC: Thank you for catching that. It is an error and has been corrected to read "ion fraction".**

**References:**

Cappa, C. D., Williams, E. J., Lack, D. A., Buffaloe, G. M., Coffman, D., Hayden, K. L., Herndon, S. C., Lerner, B. M., Li, S. M., Massoli, P., McLaren, R., Nuaaman, I., Onasch, T. B., and Quinn, P. K.: A case study into the measurement of ship emissions from plume intercepts of the NOAA ship Miller Freeman, Atmos. Chem. Phys., 14, 1337-1352, 10.5194/acp-14-1337-2014, 2014.

DeCarlo, P. F., Slowik, J. G., Worsnop, D. R., Davidovits, P., and Jimenez, J. L.: Particle Morphology and Density Characterization by Combined Mobility and Aerodynamic Diameter Measurements. Part 1: Theory, Aerosol Science and Technology, 38, 1185-1205, 10.1080/027868290903907, 2004.

Drinovec, L., Močnik, G., Zotter, P., Prévôt, A. S. H., Ruckstuhl, C., Coz, E., Rupakheti, M., Sciare, J., Müller, T., Wiedensohler, A., and Hansen, A. D. A.: The "dual-spot"

Aethalometer: an improved measurement of aerosol black carbon with real-time loading compensation, Atmos. Meas. Tech., 8, 1965-1979, 10.5194/amt-8-1965-2015, 2015.

Lin, C., Ceburnis, D., Trubetskaya, A., Xu, W., Smith, W., Hellebust, S., Wenger, J., O'Dowd, C., and Ovadnevaite, J.: On the use of reference mass spectra for reducing uncertainty in source apportionment of solid-fuel burning in ambient organic aerosol, Atmos. Meas. Tech., 14, 6905-6916, 10.5194/amt-14-6905-2021, 2021.

Lin, C., Ceburnis, D., Xu, W., Heffernan, E., Hellebust, S., Gallagher, J., Huang, R. J., O'Dowd, C., and Ovadnevaite, J.: The impact of traffic on air quality in Ireland: insights from the simultaneous kerbside and suburban monitoring of submicron aerosols, Atmos. Chem. Phys., 20, 10513-10529, 10.5194/acp-20-10513-2020, 2020.

Lin, C., Ceburnis, D., Hellebust, S., Buckley, P., Wenger, J., Canonaco, F., Prévôt, A. S. H., Huang, R.-J., O'Dowd, C., and Ovadnevaite, J.: Characterization of Primary Organic Aerosol from Domestic Wood, Peat, and Coal Burning in Ireland, Environmental Science & Technology, 51, 10624-10632, 10.1021/acs.est.7b01926, 2017.

Port Directive (EU) 2016/802 of the European Parliament and of the Council of 11 May 2016 relating to a reduction in the sulphur content of certain liquid fuels [2016] OJ L 132/58.

Trubetskaya, A., Lin, C., Ovadnevaite, J., Ceburnis, D., O'Dowd, C., Leahy, J. J., Monaghan, R. F. D., Johnson, R., Layden, P., and Smith, W.: Study of Emissions from Domestic Solid-Fuel Stove Combustion in Ireland, Energy & Fuels, 35, 4966-4978, 10.1021/acs.energyfuels.0c04148, 2021.

Wu, Y., Liu, D., Wang, X., Li, S., Zhang, J., Qiu, H., Ding, S., Hu, K., Li, W., Tian, P., Liu, Q., Zhao, D., Ma, E., Chen, M., Xu, H., Ouyang, B., Chen, Y., Kong, S., Ge, X., and Liu, H.: Ambient marine shipping emissions determined by vessel operation mode along the East China Sea, Science of The Total Environment, 769, 144713, https://doi.org/10.1016/j.scitotenv.2020.144713, 2021.

Yu, C., Pasternak, D., Lee, J., Yang, M., Bell, T., Bower, K., Wu, H., Liu, D., Reed, C., Bauguitte, S. and Cliff, S.: Characterizing the particle composition and cloud condensation nuclei from shipping emission in Western Europe, Environmental Science & Technology, 54, 24, 2020.

Yuan, T., Song, H., Oreopoulos, L., Wood, R., Bian, H., Breen, K., Chin, M., Yu, H., Barahona, D., Meyer, K., and Platnick, S.: Abrupt reduction in shipping emission as an inadvertent geoengineering termination shock produces substantial radiative warming, Communications Earth & Environment, 5, 281, 10.1038/s43247-024-01442-3, 2024.